# Enhancing Pathogen Detection in Implant-Related Infections through Chemical Antibiofilm Strategies: A Comprehensive Review

**DOI:** 10.3390/antibiotics13070678

**Published:** 2024-07-22

**Authors:** Fabiana Giarritiello, Carlo Luca Romanò, Guenter Lob, Joseph Benevenia, Hiroyuki Tsuchiya, Emanuele Zappia, Lorenzo Drago

**Affiliations:** 1Department of Medicine and Health Sciences “V. Tiberio”, University of Molise, 86100 Campobasso, Italy; f.giarritiello@studenti.unimol.it (F.G.); e.zappia@studenti.unimol.it (E.Z.); 2Romano Institute, 1001 Tirana, Albania; info@romanoinstitute.com; 3Section Injury Prevention, Deutsche Gesellschaft für Orthopädie und Unfallchirurgie (DGOU), 10117 Berlin, Germany; prof.lob@medlob.de; 4Orthopaedics Department, Rutgers New Jersey Medical School, Newark, NJ 07103, USA; benevejo@njms.rutgers.edu; 5Department of Orthopaedic Surgery, Graduate School of Medical Science, Kanazawa University, Kanazawa 921-8641, Japan; tsuchi@med.kanazawa-u.ac.jp; 6Clinical Microbiology and Microbiome Laboratory, Department of Biomedical Sciences for Health, University of Milan, 20133 Milan, Italy; 7UOC Laboratory of Clinical Medicine with Specialized Areas, IRCCS MultiMedica, 20138 Milan, Italy

**Keywords:** infections, biofilm, diagnostic techniques, chemical agents, implant-related infections (IRIs), orthopedic and cardiovascular devices, revision surgery

## Abstract

Implant-related infections (IRIs) represent a significant challenge to modern surgery. The occurrence of these infections is due to the ability of pathogens to aggregate and form biofilms, which presents a challenge to both the diagnosis and subsequent treatment of the infection. Biofilms provide pathogens with protection from the host immune response and antibiotics, making detection difficult and complicating both single-stage and two-stage revision procedures. This narrative review examines advanced chemical antibiofilm techniques with the aim of improving the detection and identification of pathogens in IRIs. The articles included in this review were selected from databases such as PubMed, Scopus, MDPI and SpringerLink, which focus on recent studies evaluating the efficacy and enhanced accuracy of microbiological sampling and culture following the use of chemical antibiofilm. Although promising results have been achieved with the successful application of some antibiofilm chemical pre-treatment methods, mainly in orthopedics and in cardiovascular surgery, further research is required to optimize and expand their routine use in the clinical setting. This is necessary to ensure their safety, efficacy and integration into diagnostic protocols. Future studies should focus on standardizing these techniques and evaluating their effectiveness in large-scale clinical trials. This review emphasizes the importance of interdisciplinary collaboration in developing reliable diagnostic tools and highlights the need for innovative approaches to improve outcomes for patients undergoing both single-stage and two-stage revision surgery for implant-related infections.

## 1. Introduction

Implant-related infections (IRIs) represent one of the most significant challenges in modern surgery, with an increasing prevalence due to the rise in joint replacement surgeries and the widespread use of implantable medical devices [1,2]. When an implant is inserted, the human body identifies it as a foreign object, establishing a physiological balance between the host (the human body) and the implant. This phenomenon, known as biocompatibility, can be severely compromised if bacteria adhere to the surface of the implant, potentially leading to a form of implant rejection [3]. IRIs are often caused by pathogenic microorganisms that have the ability to adhere to implant surfaces, aggregate, and form biofilms [4,5,6,7,8,9,10]. The accurate diagnosis of IRIs is crucial for successful treatment. Misdiagnosis can lead to inappropriate treatments that fail to eradicate pathogens, resulting in repeated surgical interventions and an increased risk of complications for the patient. Bacterial biofilms make the diagnosis of IRIs particularly challenging, as bacteria encapsulated within the biofilm often evade traditional diagnostic methods [11]. This leads to standard microbiological cultures returning false-negative results, further delaying appropriate treatment [12]. Advanced chemical antibiofilm strategies aim to disrupt these protective structures, thereby improving the detection and identification of pathogens in IRIs [13]. The adoption of these methods can significantly enhance the sensitivity of diagnostic tests, allowing for more accurate bacterial detection and better guiding treatment decisions. Additionally, the review will include a section on innovative antibiofilm treatment techniques for IRIs.

## 2. Single-Stage and Two-Stage Revision Procedures in Orthopedics

Implant-related infections caused by biofilms represent a significant challenge in surgery, necessitating the evolution of revision procedures for effective treatment.

Traditionally, two-stage revision has been the gold standard for the treatment of chronic infections, particularly in the context of orthopedic prosthetic infections [14,15,16,17,18,19,20]. This approach involves removing the infected prosthesis, inserting an antibiotic-loaded spacer, administering systemic antibiotic therapy, and eventually replacing the prosthesis. The success rate for two-stage revisions is reported to range from 60% to 100% [21,22,23,24,25,26]. Despite its effectiveness, this technique has several disadvantages, including high costs, prolonged immobilization, joint stiffness, pain, and the requirement for multiple surgeries [27,28]. Prolonged targeted systemic antibiotic administration and eventually the local antimicrobial protection of the implant are commonly reported as a required part of the patient’s management [29].

On the other hand, a “one-shot” procedure, a single-stage revision, demands the removal of all infected biomaterials, an extremely meticulous debridement and the careful management of local and systemic comorbid factors. The advantages of single-stage revision include reduced costs, faster functional recovery, and fewer required surgeries. Retrospective studies indicate that single-stage revision has a success rate of 88%, comparable to the two-stage procedure [30]. Once again, pathogen identification is considered a key factor for the success, allowing for the correct choice of the local and systemic antimicrobial administration [31].

In conclusion, while both procedures have demonstrated efficacy in treating infections, the choice between single-stage and two-stage revision should be based on a thorough assessment of the patient’s clinical condition and the capabilities of the treating center. Both procedures follow fundamental principles, including removal of the infected implant, thorough debridement of infected tissues, and targeted antibiotic application. This requires meticulous microbiological analysis, including correct selection and collection of samples during surgery, transportation, and processing of samples in the microbiology laboratory, preferably using anti-biofilm pre-treatment techniques [32]. The selection of samples is of paramount importance and aims to maintain the integrity and preserve the samples from cross-contamination. Additionally, this selection is always influenced by the diagnostic technique intended to be used. For instance, in the context of implant-related infections, once the site is infected, the analysis proceeds with the explanted material. Physical techniques like sonication can only analyze prosthetic materials, whereas chemical pretreatment techniques can be applied to both tissues and biological materials. Therefore, the samples chosen for microbiological analysis are those best suited to each diagnostic technique and must be free from contamination. These samples are extracted from the patient and placed directly into sterile containers designated for transport and microbiological analysis [33].

## 3. Heart Valve Prosthesis Infections

Heart valve prosthesis infections, particularly prosthetic valve endocarditis (PVE), are significant contributors to morbidity and mortality in cardiovascular surgery. The overall incidence of PVE has been reported to be between 0.3% and 1.2% per patient/year, with mortality rates reaching up to 50% [34,35,36]. PVE accounts for 20% of all cases of infective endocarditis (IE), with an increasing incidence. The timing of infection often reflects different pathogenic mechanisms, and PVE is typically classified into early-onset and late-onset groups based on the time elapsed from surgery.

Traditional culture methods for diagnosing native valve (NV) and prosthetic valve (PV) infections often fall short in effectiveness. Despite advancements in microbiological technologies, no single technique has emerged as a definitive reference standard. Blood culture remains the cornerstone for the diagnosis of IE; however, blood culture–negative IE can occur in up to 31% of all cases, posing diagnostic and therapeutic dilemmas [37,38].

## 4. Diagnostic Challenges Posed by Biofilm in Implant-Related Infections

The diagnosis of implant-related infections (IRIs) is significantly complicated by the presence of biofilm. Biofilms are complex structures composed of bacterial communities encapsulated in an extracellular polymeric substance (EPS) matrix that adheres to implant surfaces [7]. This matrix provides a protective environment for bacteria, shielding them from the host immune system and antibiotics [8,9,10]. The lifecycle of a biofilm results in its maturation into a complex structure, which can then detach bacterial cells, allowing them to colonize new surfaces, as summarized in Figure 1. Consequently, bacteria within biofilms often evade traditional diagnostic methods, leading to false-negative results in standard microbiological cultures. This diagnostic challenge is critical, as a delayed or incorrect diagnosis can result in the persistence of infection and the need for revision surgery [39].

Traditional diagnostic approaches, including tissue sampling and culture, frequently fail to detect biofilm-encased bacteria due to their resilient nature. These methods typically yield low sensitivity and specificity when biofilms are involved, causing clinicians to miss the underlying infection [40,41]. Furthermore, biofilms can produce small colony variants and dormant cells that are particularly difficult to detect and may not grow in standard culture conditions [42,43,44,45].

Numerous techniques have been proposed to improve the sensitivity and specificity of traditional bacterial cultures. Among these, sonication, a physical approach that uses sound waves to break down the extracellular matrix of the biofilm, has shown promising results [46]. The sonication technique involves the use of an ultrasonic bath or a probe sonicator to emit high-frequency sound waves, which disintegrate the biofilm and release the encapsulated bacteria. Samples are typically immersed in an ultrasonic bath for a duration ranging from 5 to 15 min, depending on the specific protocol used. However, due to the resilience of bacterial behavior, technical dependency, and costs, sonication is becoming obsolete and less commonly used. This is because it presents several disadvantages, such as the possibility of destroying microorganisms during the process, thereby reducing their viability and the accuracy of the resulting microbiological cultures; the inability to process cement and tissue samples; cross-contamination; and the requirement for specific equipment and containers for transport and processing, making it expensive [47,48]. Since sonication is primarily suitable for prosthetic materials, it can be useful for certain types of samples, but its limitations reduce its overall applicability in diagnostic procedures.

Molecular techniques such as polymerase chain reaction (PCR) offer high sensitivity and specificity, allowing for the identification of bacterial DNA even in small quantities [49,50]. However, the advent of next-generation sequencing (NGS) technologies, including 16S rRNA amplicon sequencing, shotgun metagenomics, and metatranscriptomics, is revolutionizing pathogen detection. Multiplex PCR and 16S rRNA sequencing have demonstrated improved sensitivity compared to traditional cultures [51,52,53,54,55,56,57,58,59,60,61], although the dependence of PCR on specific primers limits its range [54,55]. NGS can detect a wide range of organisms, including non-culturable and non-viable ones, with metagenomic NGS providing detailed genomic data and metatranscriptomic NGS offering insights into active infections [62,63,64,65,66,67,68,69].

With technological advancements in microbiology, promising chemical approaches are gaining traction. These approaches demonstrate increases in diagnostic sensitivity and specificity comparable to traditional techniques but are less costly, less technically dependent, and more accessible to all hospital and diagnostic centers [70].

**Figure 1 antibiotics-13-00678-f001:**
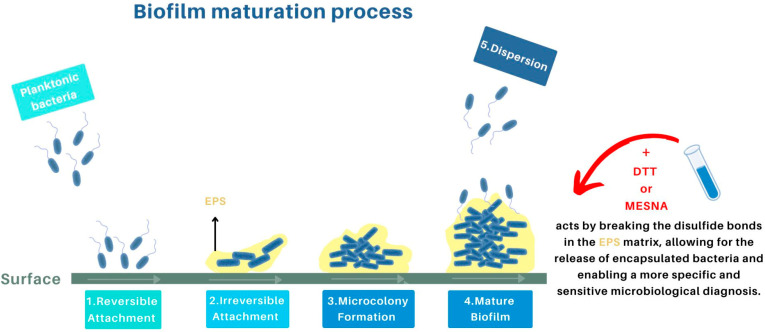
Biofilm formation process and how chemical treatments act when added to biofilm-infected sample [71,72,73,74,75,76]. Biofilm formation process and maturation: pathogens encapsulated in EPS (Extracellular Polymeric Substances) shown in yellow that adheres to implant surfaces; the effects of antibiofilm pre-treatment to counteract it, with possible chemical treatments highlighted in red: DTT (Dithiothreitol) and MESNA (2-Mercaptoethane sulfonate sodium) as described in Section 5.1.

## 5. Recent Antibiofilm Techniques

Given the significant challenges posed by microbial biofilm to identify recent advancements in chemical antibiofilm techniques aimed at improving pathogen detection in IRIs, a comprehensive review was conducted. Articles were selected from databases such as PubMed, Scopus, MDPI and SpringerLink, focusing on studies published between January 2018 and June 2024. The search terms used included “chemical antibiofilm”, “microbiological sampling”, “pathogen detection”, and “culture accuracy”. Only studies that evaluated the efficacy and enhanced accuracy of microbiological sampling and culture following the use of chemical antibiofilm agents were included in this review.

The initial research yielded numerous articles related to current treatments for implant-related infections, employing both chemical and non-chemical methods. By refining our search, we were able to focus specifically on the diagnostic aspect, which is the primary focus of this review. However, the findings from all the articles retrieved in our search were significant. Therefore, we decided to divide the discussion section into two parts. The first part concentrates on the chemical techniques used as antibiofilm methods that precede traditional microbiological cultures, enhancing and refining bacterial counts. The second part addresses emerging treatments for IRIs to improve clinical outcomes.

Both aspects are fundamental and essential for providing a comprehensive, up-to-date overview of the approaches to IRIs.

### 5.1. Diagnostic Advances Due to the Use of Chemical Agents in IRIs

In this first section, we analyze recent studies on chemical agents that have been identified over time to possess antibiofilm properties. In this section, we analyze recent studies on chemical agents that have been identified over time for their antibiofilm properties. These include DTT, as well as chelating and reducing agents such as DIGEST-EUR® and MESNA. These promising chemical agents are used in the pre-treatment of biofilm-encased bacteria to enhance the sensitivity and specificity of microbiological cultures without affecting microbial viability.

#### 5.1.1. Dithiothreitol (DTT)

Dithiothreitol (DTT) has emerged as a promising chemical agent used in the pre-treatment of biofilm-encased bacteria to enhance the sensitivity and specificity of microbiological cultures without affecting microbial viability. It works by breaking disulfide bonds in the extracellular polymeric substance (EPS) matrix of biofilms, effectively releasing the bacteria encapsulated within.

The chemical process involves treating samples with a 0.1% DTT solution, typically at a concentration of 25 mM for 15 min. This treatment increases the likelihood of detecting these bacteria in subsequent cultures. Studies have demonstrated that DTT treatment significantly improves the detection rates of biofilm-associated bacteria compared to untreated samples. The use of DTT in clinical settings has been associated with higher diagnostic accuracy, making it a valuable tool in the management of implant-related infections (IRIs) [71].

Another study highlights the efficacy of DIGEST-EUR^®^, a mucolytic agent used for rapid digestion and mucus fluidification. It has been shown that DIGEST-EUR^®^ works by breaking disulfide bonds because it also contains DTT. The chemical procedure involves treating samples with a DIGEST-EUR^®^ solution for 15 min at 37 °C [72].

This underscores the ability of DTT and its various formulations to disrupt biofilms before classical analysis, supporting its use in enhancing diagnostic accuracy.

In line with these findings, a recently developed diagnostic device, MicroDTTect, represents a significant advancement in infection management. This innovative device contains DTT and operates within a fully closed system, facilitating the collection, transport, and processing of samples directly from the operating room to the microbiology laboratory. This system ensures the preservation of sample integrity and prevents any form of contamination throughout the entire analytical process [73,74].

#### 5.1.2. Chelating Agent Ethylenediaminetetraacetic Acid (EDTA)

Ethylenediaminetetraacetic acid (EDTA) is a chelating agent that has shown promise in the dislodgement of biofilm-encased bacteria. EDTA works by chelating divalent cations such as calcium, magnesium, zinc, and iron, which are essential for the stability of the extracellular polymeric substance (EPS) matrix of biofilms. By sequestering these cations, EDTA destabilizes the biofilm structure, facilitating the release of bacteria encapsulated within.

The samples are treated with an EDTA solution, typically at a concentration of 25 mM for 15 min. This treatment effectively disrupts the biofilm matrix by chelating the necessary cations. The treated samples are then subjected to standard microbiological culturing techniques. Studies have shown that EDTA treatment significantly improves the detection rates of biofilm-associated bacteria compared to untreated samples. The use of EDTA in clinical settings has been associated with higher diagnostic accuracy, making it a valuable tool in the management of implant-related infections (IRIs). EDTA has been found to be particularly effective in dislodging biofilms of various bacterial strains, including *Staphylococcus epidermidis*, *Staphylococcus aureus*, *Escherichia coli*, and *Pseudomonas aeruginosa* [75].

#### 5.1.3. MESNA

Sodium 2-mercaptoethanesulfonate (MESNA) is a synthetic thiol compound that has shown promise in the dislodgement of biofilm-encased bacteria, enhancing the accuracy of microbiological cultures. MESNA works by breaking disulfide bonds in the extracellular polymeric substance (EPS) matrix of biofilms, effectively releasing the bacteria encapsulated within. Its application in pediatric cholesteatoma has demonstrated significant changes in biofilm structure, aiding in the management of chronic infections. Samples are treated with a 4% MESNA solution for 10 min at 37 °C. This treatment breaks down the biofilm matrix by breaking disulfide bonds. The treated samples are then subjected to standard microbiological culturing techniques. MESNA has been found to be particularly effective in dislodging biofilms of various bacterial strains, including *Staphylococcus epidermidis*, *Staphylococcus aureus*, *Escherichia coli*, and *Pseudomonas aeruginosa* [77] (Table 1).

### 5.2. Innovative Treatment Approaches for Implant-Related Infections

The second part of this review explores advanced chemical techniques for the breakdown of biofilms, aimed at improving the treatment of IRIs.

Treatment techniques resulting from the above type of research include:Antimicrobial Peptides (AMP): These peptides are used to prevent the formation of biofilms on surfaces of polymethyl methacrylate (PMMA). They exhibit antibacterial activity by breaking the cell membrane of pathogens, reducing bacterial adhesion and biofilm formation [78].AIEgen: These probes, employed as theranostic instruments, enhance penetration into biofilms and facilitate both the diagnosis and treatment of multi-resistant bacterial biofilm infections due to their capacity to generate localized heat [79].Cold Plasma Treatments: By modifying the surfaces of materials, cold plasma treatments render surfaces anti-adhesive and antibacterial, preventing adhesion and bacterial growth. This treatment is designed to modify the surface of medical implants and counteract the adhesion phase of biofilm formation, thereby improving the antibacterial activity of the materials [76].Silver Nanoparticles (AgNPs): Derived from Lactobacillus casei, these nanoparticles possess powerful antibiofilm properties, inhibiting the formation and growth of bacterial biofilms. These nanocubes are activated by NIR light to generate heat, which destroys bacterial biofilms and improves the effectiveness of antibiotics. They offer a promising approach to the treatment of implant-related infections due to their ability to selectively degrade biofilms [80].

Selected articles explore various innovative approaches, each with unique mechanisms of action and proven efficacy against bacterial biofilms. Below is a summary of the methodologies used and their effectiveness in treating IRIs.

The following Table 2 summarizes the methods, year of publication, efficacy on IRIs, and mechanisms of action of the selected articles:

## 6. Discussion and Conclusions

This review highlights the critical role of chemical antibiofilm agents in both the diagnosis and treatment of implant-related infections (IRIs). Techniques such as DTT, EDTA, and MESNA have demonstrated significant promise in disrupting biofilms and detecting encapsulated bacteria within the EPS matrix. These advancements have improved diagnostic accuracy and treatment outcomes across various surgical fields, particularly in cardiovascular surgery. The application of these agents has proven effective in managing implant-related infections, reducing complications, and improving patient prognosis.

Following diagnostic techniques, recent emerging treatments using nanoparticles have also shown substantial benefits.

Nowadays, the frequent use of antibiotics for treating biofilm-associated infections has led to a steady and progressive increase in antibiotic-resistant bacteria that cannot be eradicated with traditional antibiotic treatments. Bacterial biofilms are the leading cause of healthcare-associated infections in humans. Therefore, there is an urgent need for new approaches and strategies to inhibit biofilm formation [81].

The integration of chemical antibiofilm agents into clinical practice has provided a new dimension to the management of IRIs. However, challenges remain concerning the standardization of these techniques, cost considerations, and the need for specialized equipment. Continued research and clinical trials are essential to optimize these methods and ensure their widespread adoption.

## 7. Future Needs to Improve the Surgical Revision Procedure

Biofilms present a significant challenge in the context of implant-related infections (IRIs), particularly affecting the success of single-stage and two-stage revision procedures [82]. Various efforts are underway to identify new candidates for preventing and treating biofilm-forming bacteria. Chemical agents, along with nanotechnology-based methods, have shown promise in antibiofilm activity. However, their effectiveness in clinical settings is often limited due to a lack of understanding of the underlying mechanisms.

A major challenge lies in bridging the gap between in vitro and in vivo testing. Despite promising in vitro results, current models fail to replicate the complex conditions of the human body, where biofilm formation occurs on implants. This limitation underscores the need for more realistic models to better predict clinical outcomes. Developing standardized protocols for the use of chemical antibiofilm agents during single-stage and two-stage revision procedures is crucial. The consistent application of these protocols, based on robust clinical evidence, will improve patient outcomes by ensuring that the most effective methods are used uniformly across clinical settings.

Cost reduction and continued research into more effective antibiofilm agents are necessary to facilitate broader adoption and improve treatment efficacy. Collaboration between microbiologists, surgeons, and engineers can drive innovation and develop integrated biofilm management strategies. Advanced diagnostic tools that quickly detect biofilm-associated infections will enhance early intervention and treatment success.

These findings emphasize the need for innovative strategies to improve revision procedures, ultimately enhancing patient outcomes and managing implant-related infections more effectively in clinical surgery.

## Figures and Tables

**Table 1 antibiotics-13-00678-t001:** Chemical antibiofilm agents employed to improve diagnosis of biofilm infections.

Method	Year	Method of Application	Mechanism of Action
DTT solution and formulations	2022 [71,72,73,74]	Samples are treated with a 0.1% DTT solution at a concentration of 25 mM for 15 min.	Breaks disulfide bonds in EPS matrix, releasing encapsulated bacteria
EDTA	2020 [75]	Samples are treated with an EDTA solution at a concentration of 25 mM for 15 min.	Chelates divalent cations essential for EPS stability, destabilizing biofilm structure.
MESNA	2018 [77]	Samples are treated with a 4% MESNA solution for 10 min at 37 °C.	Breaks disulfide bonds in EPS matrix, releasing encapsulated bacteria

**Table 2 antibiotics-13-00678-t002:** Recent promising strategies for counteracting biofilm formation.

Method	Years	Effectiveness on the Treatment of IRIs	Mechanism of Action
Antimicrobial Peptides (AMP)	2019 [78]	Significant reduction in bacterial adhesion and biofilm formation	Disrupt bacterial cell membranes, preventing adhesion and biofilm formation
AIEgen	2022 [79]	Effective biofilm penetration and increased antibacterial efficacy against multidrug-resistant bacteria	Enhance penetration and disrupt biofilms via photothermal effect
Cold Plasma Treatments	2021 [76]	Significant reduction in bacterial adhesion and biofilm formation on various surfaces	Modify surface properties to reduce bacterial adhesion and biofilm formation
Silver Nanoparticles (AgNPs)	2023 [80]	Effective inhibition of biofilm formation and growth of bacterial biofilms	Interact with bacterial cell membranes to disrupt biofilm formation and bacterial growth

## Data Availability

No new data were created or analyzed in this study.

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
