# Peer review of "Enhancing Pathogen Detection in Implant-Related Infections through Chemical Antibiofilm Strategies: A Comprehensive Review"

_antibiotics, 2024, doi:10.3390/antibiotics13070678_

Round 1

Reviewer 1 Report

Comments and Suggestions for Authors

I would like to extend my congratulations to the authors for this excellent review. The article is concise and well written.

The manuscript addresses the use of chemical-based strategies to mitigate the impact of biofilm formation in the diagnosis and treatment of implant-related infections. However, I believe that some sections need to be implemented in order to provide a more comprehensive overview.

Section 2. Single-Stage and Two-Stage Revision Procedures in Orthopaedics:

Line 87: Please define the clinical specimen and the clinical specimen of choice, including the collection procedure.

Section 3. Heart Valves Prosthesis Infections:

Lines 102-110; This sentence fits in section 5.1., subsection Dithiothreitol (DTT); the technique is described here.

Section 4. Diagnostic Challenges Posed by Biofilm in Implant-Related Infections:

Lines 120-124; More detail of the traditional method and a reasoned explanation of the origin of the diagnostic failure is required.

Lines 123-124; Please insert bibliographic citation. The scientific literature does not make this clear.

Line 127; Sonication of specimens is the method of choice for diagnosis of intravascular catheter-related bloodstream infection. Please explain in detail this technique, equipment, sonication times, frequencies, etc., false negatives, including disadvantages such as destruction of microorganisms .....

Section 5.2 Treatments advances strategies to the IRIs. More detail is required in the description of the different strategies with emphasis on their potential prophylactic (preventing biofilm formation) or therapeutic (treatment of the biofilm formed) use.

Table 1; . Chemical Antibiofilm Agents Employed Against Biofilm Infections. Specify that this is for diagnostic purposes.

Minor:

Line 63; evolution or evaluation???

Line 74. Introduce the process first. Then the advantages

Linea 113-115; Check if this sentence is redundant. Use the term lifecycle.

Linea 134 Nevertheless, these techniques alone are not sufficient to achieve accurate bacterial counts; (Real time PCR????); citation is required

Author Response

Dear Reviewer 1,

We would like to extend our sincere thanks for your thorough review and for the positive feedback on our manuscript. We appreciate your insightful comments and suggestions, which have undoubtedly contributed to enhancing the clarity and comprehensiveness of our review. Below, we address the specific points you raised and describe the changes made.

MAJOR CORRECTIONS:

Comment 1:

  • “Section 2. Single-Stage and Two-Stage Revision Procedures in Orthopaedics:

Line 87: Please define the clinical specimen and the clinical specimen of choice, including the collection procedure.”

Response 1:

  • We appreciated the suggestion and made changes to the text to better specify the chosen clinical sample and the collection procedure. These changes are reflected in the revised manuscript on Line 89-99.

Additionally, we have included the following reference:

[33]:Janz, V.; Wassilew, G.I.; Hasart, O.; Tohtz, S.; Perka, C. Improvement in the detection rate of PJI in total hip arthroplasty through multiple sonicate fluid cultures. *J. Orthop. Res.* 2013, *31*, 2021–2024.

Comment 2:

“Section 3. Heart Valves Prosthesis Infections: 

  • Lines 102-110; This sentence fits in section 5.1., subsection Dithiothreitol (DTT); the technique is described here.”

Response 2:

  • We agree with your suggestion and have removed this paragraph from the section 3; along with the associated reference [old reference 39].

Comment 3:

“Section 4. Diagnostic Challenges Posed by Biofilm in Implant-Related Infections: 

  • Lines 120-124; More detail of the traditional method and a reasoned explanation of the origin of the diagnostic failure is required.

  • Lines 123-124; Please insert bibliographic citation. The scientific literature does not make this clear.

  • Line 127; Sonication of specimens is the method of choice for diagnosis of intravascular catheter-related bloodstream infection. Please explain in detail this technique, equipment, sonication times, frequencies, etc., false negatives, including disadvantages such as destruction of microorganisms .....”

Response 3:

  • We thank you for this correction. We believe that the diagnostic failure mentioned in lines 120-124 is due to the difficulty of detecting pathogens within the biofilm, as further explained in lines 115 - 119 of the same section; so, we don’t have add more text to avoid redundancy. (Line of revised manuscript: 130-133)

  • Thank you for this correction. We have added references to support this statement (Line of revised manuscript 133-135). The added references are [42-45] in the revised manuscript :
  1. Brown, M.R.W.; Allison, D.G.; Gilbert, P. Resistance of bacterial biofilms to antibiotics: a growth-rate related effect? *J. Antimicrob. Chemother.* 1988, *22*, 777–783.
  2. Lewis, K. Riddle of biofilm resistance. *Antimicrob. Agents Chemother.* 2001, *45*, 999–1007.
  3. Stewart, P.S.; Franklin, M.J. Physiological heterogeneity in biofilms. *Nat. Rev. Microbiol.* 2008, *6*, 199–210.
  4. Kim, L. Persister cells, dormancy and infectious disease. *Nat. Rev. Microbiol.* 2007, *5*, 48–56.

  • Thank you for this important suggestion. We agree and have added a detailed explanation of sonication procedures, materials, techniques, and their disadvantages, such as the destruction of microorganisms. These details can be found in the revised manuscript on lines 139-151 with the references [47-48] reorganized accordingly:
  1. Romanò, C.L.; Scarponi, S.; Gallazzi, E.; Romanò, D.; Drago, L. Cost-Benefit Analysis of Antibiofilm Microbiological Techniques for Peri-Prosthetic Joint Infection Diagnosis. *J. Clin. Med.* 2020, *9*(7), 2203.
  2. Sambri, A.; Cadossi, M.; Giannini, S.; Pignatti, G.; Marcacci, M.; Ferrari, M.C.; Donati, D.M.; De Paolis, M. Is Treatment with Dithiothreitol More Effective than Sonication for the Diagnosis of Prosthetic Joint Infection? *Clin. Orthop. Relat. Res.* 2018, *476*, 137–145.

Comment 4:

“Section 5.2 Treatments advances strategies to the IRIs.

More detail is required in the description of the different strategies with emphasis on their potential prophylactic (preventing biofilm formation) or therapeutic (treatment of the biofilm formed) use. 

-Table 1: Chemical Antibiofilm Agents Employed Against Biofilm Infections. Specify that this is for diagnostic purposes.”

Response 4:

  • Section 5.2: Thank you for your insightful comment. While we appreciate and understand your suggestion for a more detailed description of the treatment methods and examples of their clinical or pre-clinical applications, we would like to clarify that our primary focus is on chemical antibiofilm agents for the diagnosis of implant-related infections (IRIs). The Treatments section was included because our search keywords also yielded numerous novel treatment techniques. Although this section is not our primary focus, it is included to provide a comprehensive overview. We have chosen not to explore these techniques in greater detail to avoid redundancy with other published reviews that have thoroughly examined these topics. We believe that maintaining our focus on diagnostic methods will provide a clearer and more concise contribution to the field. 
  • Table 1: We agree with the reviewer and have changed the table description to: “Table 1. Chemical Antibiofilm Agents Employed to Improve Diagnosis of Biofilm Infections.”

MINOR CORRECTIONS:

  • Line 63: The correct word is "evolution," and this has been updated.
  • Line 74: We agree and have restructured the text to introduce the process first, followed by the advantages. These changes can be found on lines 74-79 of revised manuscript.
  • Lines 113-115: Thanks! We have revised this section to eliminate redundancy and used the term "lifecycle." The changes are reflected on lines 123-125.
  • Line 134: We agree with the reviewer and have rephrased this sentence for clarity. The revised text can be found on lines 152-158 of revised manuscript, with no new references required.

We sincerely thank the reviewer for these valuable corrections, which have significantly improved our manuscript. The revisions are highlighted in the revised manuscript to facilitate your review

Best regards,

Authors of “antibiotics-3102797”.

Reviewer 2 Report

Comments and Suggestions for Authors

The work concerns the clinically important topic of implant-related infections and I rate it highly. However, to increase its cognitive value, I would suggest adding more detailed information. Section 5.2 Treatments advances strategies to the IRIs lacks a broader description of the mentioned methods and examples of clinical or pre-clinical application of the methods listed in Table 2. This part should be supplemented and a detailed mechanism of action should be added, e.g. in the form of a diagram regarding the effects on biofilm.

Moreover, I suggest to add literature references of the cited content to tables 1 and 2, maybe in the same column as “Year”.

Author Response

Dear Reviewer 2,

We would like to extend our sincere thanks for your positive feedback and valuable suggestions regarding our manuscript. We appreciate your insightful comments and have carefully considered your recommendations.

MAJOR CORRECTIONS:

Comment 1:

  • “The work concerns the clinically important topic of implant-related infections and I rate it highly. However, to increase its cognitive value, I would suggest adding more detailed information. Section 5.2 Treatments advances strategies to the IRIs lacks a broader description of the mentioned methods and examples of clinical or pre-clinical application of the methods listed in Table 2. This part should be supplemented and a detailed mechanism of action should be added, e.g. in the form of a diagram regarding the effects on biofilm.

Response 1:

  • Thank you for your insightful comment. While we appreciate and understand your suggestion for a more detailed description of the treatment methods and examples of their clinical or pre-clinical applications, we would like to clarify that our primary focus is on chemical antibiofilm agents for the diagnosis of implant-related infections (IRIs). The Treatments section was included because our search keywords also yielded numerous novel treatment techniques. Although this section is not our primary focus, it is included to provide a comprehensive overview. We have chosen not to explore these techniques in greater detail to avoid redundancy with other published reviews that have thoroughly examined these topics. We believe that maintaining our focus on diagnostic methods will provide a clearer and more concise contribution to the field.

Comment 2:

  • Moreover, I suggest to add literature references of the cited content to tables 1 and 2, maybe in the same column as “Year”.

Response 2:

  • We found your suggestion to add literature references in the tables very useful. Therefore, we have included the relevant references in Table 1 and Table 2, in the same column as “Year.”

We sincerely thank the reviewer for these valuable corrections, which have significantly improved our manuscript. The revisions are highlighted in the revised manuscript to facilitate your review.

Best regards,

Authors of “antibiotics-3102797”.

Reviewer 3 Report

Comments and Suggestions for Authors

This is a useful paper and a good summary of implant related infections and their challenges. The glow of logic is clear. 

The following minor revisions are needed.

Line 88 - this should read "sample transport" not "samples transport"  

Line 101 - this should read "for diagnosis of IE"

Line 124 -  Suggest mention of VBNC in this context of organisms that will not be recovered using standard culture-based methods

Line 237 Table 1 - row 1 "udu15 minutes" delete udu

Line 239 "Treatments advances strategies to the IRIs" Please reword this title for clarity.

Line 240 This should read "The second part of this review explores..."

Line 253 Reword "These micelles ..." so it is clear what micelles are being referred to, as the flow of logic seems broken - why are micelles being referred to at all?.

Line 256 Given the sizeable literature on AgNPs it is not clear why just this one study was chosen, so please add an explanation. Was this the only study in the timeframe  that mentioned IRI?

Many references have the wrong formatting. The authors need to convert their Endnote references to plain text and then manually edit each reference to the correct style for the journal.  As a template, use the style of ref 42 as that one is correct.

Comments on the Quality of English Language

No concerns

Author Response

Dear Reviewer 3,

We would like to extend our sincere thanks for your positive feedback and for acknowledging the value and clarity of our manuscript. We appreciate your detailed suggestions for improvement and have implemented the necessary revisions accordingly.

The following minor revisions, as you suggest, are needed: 

  • Comments: Line 88 - this should read "sample transport" not "samples transport" 
  • Response: Thaks, done! Changed to "sample transport."- Line 88 of reviewed manuscript.

  • Comments: Line 101 - this should read "for diagnosis of IE"
  • Response: Thaks, done! Line 111 of reviewed manuscript: Corrected to "for diagnosis of IE."

  • Comments: Line 124 - Suggest mention of VBNC in this context of organisms that will not be recovered using standard culture-based methods
  • Response: Line 133 of reviewed manuscript: Thaks, done! This part was revised and justified differently.

  • Comments: Line 237 Table 1 - row 1 "udu15 minutes" delete udu
  • Response: Line 258 of reviewed manuscript, Table 1: Thaks, done! Corrected to "15 minutes.”.

  • Comments: Line 239 "Treatments advances strategies to the IRIs" Please reword this title for clarity.
  • Response: Line 260 of reviewed manuscript: Thaks, done! Reworded to “Innovative Treatment Approaches for Implant-Related Infections.”

  • Comments: Line 240 This should read "The second part of this review explores..."
  • Response: Line 261 of reviewed manuscript: Thaks, done! Changed to “The second part of this review explores…”

  • Comments: Line 253 Reword "These micelles ..." so it is clear what micelles are being referred to, as the flow of logic seems broken - why are micelles being referred to at all?
  • Response: Line 274 of reviewed manuscript: Thaks, done! Reworded for clarity.

  • Comments: Line 256 Given the sizeable literature on AgNPs it is not clear why just this one study was chosen, so please add an explanation. Was this the only study in the timeframe that mentioned IRI?
  • Response: Line 277 of reviewed manuscript: Thank you for this comment. The studies and references included in the section on treatments against biofilm formation responsible for IRIs are supplementary studies selected because they appeared in our keyword search, even though they are not part of the main topic of our review, which focuses on diagnostics rather than treatment. We included a single study for each type of existing treatment to provide a comprehensive overview of the field. However, since there are already reviews in the literature that have extensively covered the novel techniques to counteract biofilm formation, we decided to include these references but not to delve deeply into them to avoid redundancy and staying off-topic.

Additionally, we have reformatted the references to align with the style of reference 42, as you suggested.

We sincerely thank you for these valuable corrections, which have significantly improved our manuscript. The revisions are highlighted in the revised manuscript to facilitate your review.

Best regards,

Authors of “antibiotics-3102797”.

Round 2

Reviewer 1 Report

Comments and Suggestions for Authors

It is certainly a good job. Congratulations on this great review.